# SEARCH-T2I: INTERNET-AUGMENTED TEXT-TO-IMAGE GENERATION

## ABSTRACT

Current text-to-image (T2I) generation models achieve promising results, but they fail on the scenarios where the knowledge implied in the text prompt is uncertain. For example, a T2I model released in February would struggle to generate a suitable poster for a movie premiering in April, because the character designs and styles are uncertain to the model. To solve this problem, we propose an Internet-Augmented text-to-image generation framework (Search-T2I) to compel T2I models clear about such uncertain knowledge by providing them with reference images. Specifically, an active retrieval module is designed to determine whether a reference image is needed based on the given text prompt; a hierarchical image selection module is introduced to find the most suitable image returned by an image search engine to enhance the T2I model; a self-reflection mechanism is presented to continuously evaluate and refine the generated image to ensure faithful alignment with the text prompt. To evaluate the proposed framework's performance, we collect a dataset named Img-Ref-T2I, where text prompts include three types of uncertain knowledge: (1) known but rare. (2) unknown. (3) ambiguous. Moreover, we carefully craft a complex prompt to guide GPT-4o in making preference evaluation, which has been shown to have an evaluation accuracy similar to that of human preference evaluation. Experimental results demonstrate the effectiveness of our framework, outperforming GPT-4o by 30% in human evaluation.

## 1 INTRODUCTION

Text-to-image (T2I) generation models, such as Stable Diffusion Rombach et al. (2021), ControlNet Zhang et al. (2023) and FLUX Labs (2024), have attracted considerable attention for their ability to generate highly realistic images based on text prompts that often encapsulate complex and context-dependent knowledge. However, knowledge is unevenly distributed across the world, constantly evolving, and often ambiguous. These characteristics make it challenging for T2I models to perform reliably in scenarios where the knowledge implied in the text prompt is uncertain. For example, a T2I model released in February would likely struggle to generate an appropriate poster for a movie premiering in April, as it lacks access to up-to-date information about the movie. Key visual elements such as character designs, costumes, and stylistic choices may not be publicly available or finalized at the time the model was trained, which can lead to inaccurate or overly generic results. To address this issue, we propose an Internet-Augmented Text-to-Image generation framework, Search-T2I, which augments T2I models' understanding of uncertain knowledge by supplying them with relevant reference images retrieved from Internet.

We first present our overall framework for augmenting T2I models with reference images, consists of six components: active retrieval module, query generator, search engine, hierarchical image selection module, augmented T2I generation, and self-reflection mechanism, as illustrated in Figure 1. Specifically, We begin by exploring the knowledge boundaries of a T2I model to determine whether generating an accurate image for a given text prompt requires additional reference images. Next, we use large vision-language models (LVLMs) to extract queries from the text prompt and retrieve potentially useful reference images via search engines. However, directly augmenting T2I models with the retrieved images is impractical due to: (1) The number of retrieved images is typically large, making processing computationally expensive and time-consuming. (2) Although the images are ranked by relevance, those at the top are not necessarily the most suitable as reference images. To address this, we introduce a hierarchical image selection module to identify the most helpful

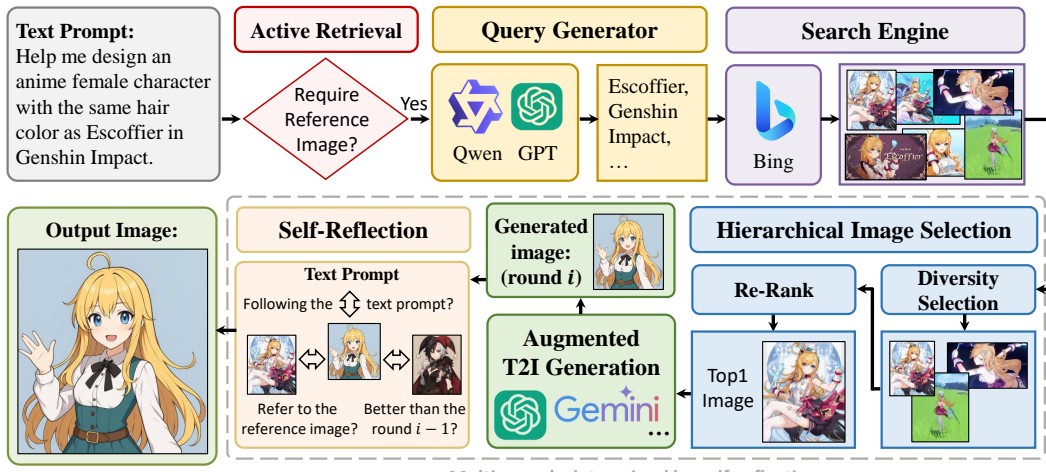

Figure 1: The proposed Search-T2I, a framework for T2I models to refer to images.

reference image for guiding image generation. It first performs an initial filtering based on diversity to form a candidate set, and then re-ranks the candidates to select the most relevant and informative reference image. Once the T2I model generates an image augmented by the selected reference image, a self-reflection mechanism is employed to evaluate the output and autonomously decide whether to reselect reference images for another generation attempt or to accept the result as final.

We then construct a dataset dubbed Img-Ref-T2I, where text prompts include three types of uncertain knowledge: (1) known but rare. (2) unknown. (3) ambiguous. *Known but rare* refers to knowledge that existed before the model's release (set as March 26, 2025, the latest update date of GPT-4o Hurst et al. (2024) in this paper) but is not commonly encountered. For example, people are generally more familiar with checkers than with Polish draughts. *Unknown* refers to new knowledge that emerges after the model's release, which is inaccessible to the model. *Ambiguous* refers to a concept that has different meanings and visual representations depending on the context. For example, drawing a map of EU member states requires specifying a point in time, as the composition of the European Union has changed over the years. Our Img-Ref-T2I dataset is designed for two tasks: general text-to-image generation and text-conditioned image editing (TI2I). To assess the quality of generated images, we design a complex prompt to perform automatic preference evaluation pipeline based on GPT-4o, which achieves evaluation accuracy comparable to human preference evaluation. Experimental results on Img-Ref-T2I validate the effectiveness of the proposed Search-T2I framework, outperforming GPT-4o by approximately 30% in human evaluations. Furthermore, experiments on two datasets: GenEval Ghosh et al. (2023) and ConceptMix Wu et al. (2024b) demonstrate that our framework does not negatively affect commonly used prompts that do not involve uncertain knowledge.

Our contributions are summarized as follows. (1) We propose Search-T2I, the first framework that integrates reference images from the Internet into T2I models, effectively mitigating inaccurate image generation caused by uncertain knowledge in text prompts. (2) We collect Img-Ref-T2I, the first dataset for evaluating the performance of T2I models under three types of scenarios where the textual knowledge is uncertain. (3) We develop a GPT-4o-based automatic preference evaluation method by prompt engineering, achieving results comparable to human preference evaluation.

## 2 RELATED WORK

### 2.1 TEXT-TO-IMAGE GENERATION

With the continuous advancement of technologies such as generative adversarial networks Goodfellow et al. (2020); Xu et al. (2018), diffusion models Song et al. (2020); Rombach et al. (2021); Zhang et al. (2023), and autoregressive models Tian et al. (2024); Sun et al. (2024a), text-to-image generation has garnered increasing attention. Works Zhang et al. (2023); Li et al. (2024b); Liu et al. (2024) emphasize the role of control signals in guiding the image generation process. Works Xie et al.

(2024); Team (2024); Wu et al. (2024a); Ge et al. (2024); Sun et al. (2025) unify generation and understanding within a single model. Works Huang et al. (2024b); Wang et al. (2023a;b;c); Sun et al. (2024b); Najdenkoska et al. (2024); Huang et al. (2024a) explore the in-context learning capabilities of text-to-image models. In contrast, we focus on retrieving helpful reference images for the image generation process to mitigate the uncertain textual knowledge, which is orthogonal to existing works. Works Blattmann et al. (2022); Sheynin et al. (2022); Chen et al. (2022); Yuan et al. (2025); Lyu et al. (2025) explore retrieving additional reference images for text-to-image generation, where the retrieval modules require model-specific training and typically retrieve from fixed, carefully curated local databases. In contrast, our proposed Search-T2I framework is training-free and retrieves reference images from the Internet, which is a constantly evolving and highly noisy source.

## 2.2 INTERNET-AUGMENTED GENERATION

Recently Internet-augmented generation (IAG) attracted increasing attention of both the natural language processing (NLP) and vision-and-language (V&L). In NLP, Komeili *et.al.* Komeili et al. (2021) demonstrate that incorporating search engines into large language models (LLMs) can reduce the generation of factually incorrect content during human dialogues. Lazaridou *et.al.* Lazaridou et al. (2022) employs few-shot prompting to allow LLMs to leverage knowledge retrieved from search engine to respond to questions involving factual and up-to-date information. Tian *et.al.* Tian et al. (2023) use IAG to build open-domain generative dialogue system for digital human. In V&L, Li *et al.* Li et al. (2024a) propose SearchLVLMs, a framework that enables existing LVLMs to access up-to-date knowledge during inference through IAG. Jiang *et al.* Jiang et al. (2024) present MMSearch to empower LVLMs for multimodal searching via IAG. Differently, we introduce IAG into the text-to-image generation task, to mitigate inaccurate image generation caused by uncertain knowledge in text prompts.

## 3 SEARCH-T2I FRAMEWORK

In this section, we introduce Search-T2I, a framework that addresses the issue brought by uncertain knowledge by integrating reference images from the Internet into them. The overview of Search-T2I is illustrated in Figure 1. For a text prompt $T$ (for TI2I, the input information additionally includes an original image $I_0$), we first use an active retrieval module to determine whether a reference image is required. Then we extract queries for $T$ and fed them into search engine to obtain potential reference images. Next, a hierarchical image selection module is used to identify the most helpful reference image, and the T2I model is augmented by the reference image to generate an output image. Finally, we employ a self-reflection mechanism to evaluate the output and determine whether to reselect reference images for another generation attempt or accept the result as final.

### 3.1 ACTIVE RETRIEVAL

To determine whether a reference image is required for $T$, it is essential to explore the knowledge boundaries of a T2I model. We attempt two different lines: (1) Judge solely based on the input information ($T$ for T2I, $T$ and $I_0$ for TI2I). (2) Judge based on both the input information and the image generated by the T2I model (without reference images). For the former, we prompt the model to analyze the input information and determine whether it contains uncertain knowledge. For the latter, we assess the instruction-following ability of the generated image to decide whether reference images are needed for regeneration. The effectiveness of active retrieval depends on the design of the prompt. Detailed experimental analysis and prompt choice are provided in Section 5.5.

### 3.2 QUERY GENERATOR

We leverage existing LVLMs, such as GPT-4o Hurst et al. (2024) and Qwen2.5-VL Bai et al. (2025), to extract queries from $T$ to obtain queries that lead search engines to return potential helpful reference images. Thanks to their language understanding capabilities, LVLMs can infer the grammatical role of each word in $T$, even when the knowledge implied in certain words is uncertain. Since some knowledge is culture-specific, we translate $T$ into two additional languages including Chinese and Japanese, and extract queries accordingly to ensure a higher recall of reference images returned

by search engines. The prompt used for guiding LVLMs in generating queries can be found in the Appendix.

## 3.3 SEARCH ENGINE

The queries in three different languages are fed separately to the search engine. By invoking the image search function, the engine directly returns multiple images. However, directly using all the returned images to enhance the T2I model is impractical. Firstly, many of the images are noisy and may mislead the T2I generation process. Additionally, the sheer number of images makes it computationally expensive and time-consuming to process them all. Although the search engine ranks the returned images based on query relevance, the top-ranked images may only match the content of the webpage they are embedded in rather than the actual visual content. Therefore, further filtering of the returned images is necessary.

## 3.4 HIERARCHICAL IMAGE SELECTION

To filter the returned images, we propose a hierarchical image selection module that performs two-step filtering, consisting of a diversity-based selection followed by a re-ranking process.

**Diversity Selection.** We first perform an initial filtering of the returned images by selecting those with the greatest diversity, aiming to ensure that the candidate set includes as many useful reference images as possible. Specifically, we extract CLIP features Radford et al. (2021) from the returned images and apply k-means clustering Jain & Dubes (1988) based on cosine similarity between these features to form N clusters. The image closest to the center of each cluster is then selected as a candidate reference image.

**Re-Rank.** The purpose of re-rank is to sort the candidate reference images and select the most helpful one. We input multiple images into LVLMs and use the prompt (can be found in the Appendix) to guide them in ranking. Notably, due to the self-reflection mechanism (described in Section 3.6), the hierarchical image selection module may be executed in multiple rounds. Therefore, we use $I_i^{ref}$ to denote the top-1 reference image selected for round $i$.

## 3.5 AUGMENTED GENERATION

After obtaining the reference image $I_i^{ref}$, We provide it during the image generation process of T2I models and indicate its role in the text prompt to perform augmented generation. The output image by augmented generation of round $i$ is denoted as $I_i^o$.

## 3.6 SELF-REFLECTION

A self-reflection mechanism is employed to evaluate the accuracy and usability of the output image $I_i^o$ generated in the current round. If the result is deemed unsatisfactory, a new round is initiated to reselect reference images and attempt generation again. The evaluation of $I_i^o$ is based on three key criteria: (1) Whether it faithfully follows the text prompt $T$. (2) Whether the reference image $I_i^{ref}$ is helpful. (3) Whether it effectively incorporates information from the reference image $I_i^{ref}$ selected in this round. (4) For $i > 1$, whether it improves upon the output image from the previous round, *i.e.*, $I_{i-1}^o$. For these four criteria, we use different prompts (provided in the Appendix) to guide GPT-4o in scoring $I_i^o$. When the total score is greater than or equal to 8, $I_i^o$ is accepted as the final output image, denoted as $I_{out}$.

## 4 IMG-REF-T2I DATASET

To evaluate the performance of T2I models in scenarios with uncertain textual knowledge, we construct a dataset named Img-Ref-T2I, which maintains high quality, as it is entirely curated and annotated by human experts. The dataset consists of a total of 240 samples: 120 for T2I and another 120 for TI2I. We categorize uncertain knowledge into three types: (1) *Known but rare*, (2) *Unknown*, and (3) *Ambiguous*. For each task's 120 samples, 30 samples correspond to each of the three uncertainty categories. In addition, we collect 30 samples that contain no uncertain knowledge, which

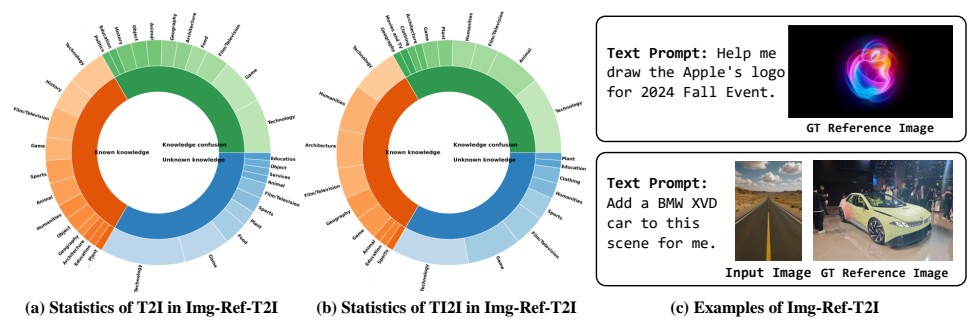

(a) Statistics of T2I in Img-Ref-T2I    (b) Statistics of TI2I in Img-Ref-T2I    (c) Examples of Img-Ref-T2I

Figure 2: The distribution and examples of the proposed Img-Ref-T2I dataset.

are used to evaluate the accuracy of the active retrieval module. *Known but rare* refers to knowledge that existed before the release date of the model. In this paper, we define this as prior to GPT-4o's most recent update, March 26, 2025. *Unknown* refers to new concepts or events that emerged after the model's release. *Ambiguous* refers to concepts that exhibit different visual representations depending on the context. For each sample containing uncertain knowledge, we manually collect a ground-truth (GT) reference image to evaluate the performance bottleneck in image generation. The distribution and examples of the Img-Ref-T2I dataset are illustrated in Figure 2.

## 5 EXPERIMENTS

### 5.1 SETTINGS

**Baselines.** We select three categories of models as baselines for our experiments: (1) T2I models, including FLUX Labs (2024) and DDPM Huberman-Spiegelglas et al. (2024). FLUX is a commonly used T2I model, while DDPM is an inverse-based T2I model capable of injecting reference image information into the generation process. (2) TI2I models, represented by Step1X-Edit Liu et al. (2025). (3) Omnipotent (Omni.) models, including Gemini-2.0-flash Gemini2 (2025) (represented by Gemini in the following text for convenience) and GPT-4o Hurst et al. (2024). These commercial models support both T2I and TI2I, and can incorporate reference images as contextual input for image generation or editing.

**Implementation Details.** For open-source models: FLUX, DDPM, and Step1X-Edit, we re-implement them via their official code repositories. For closed-source commercial models, GPT-4o and Gemini, we access the models via their official APIs. We incorporate GPT-4o and Gemini into Search-T2I, as they support reference images. We perform evaluations with a single Nvidia A100 GPU.

### 5.2 GENERATED IMAGE EVALUATION

**Text Prompt with Uncertain Knowledge.** We evaluate generated images on Img-Ref-T2I via human evaluation. For a text prompt $T$, the image generated by model $X$ is denoted as $I_{out}^X$. Each record is represented as $[T, I_{out}^X]$ (for the TI2I task, the original image $I_0$ is additionally included in the record). We ask evaluators (co-authors of this paper) to score the outputs based on three aspects: (1) Aesthetic Quality (AQ). Evaluators are asked three questions: a. (Layout) Is the layout of $I_{out}^X$ harmonious? b. (Color) Are the colors in $I_{out}^X$ coordinated? c. (Clearness) Is $I_{out}^X$ visually clear? (2) Commonsense Consistency (CC). Evaluators are asked: "Do the details in $I_{out}^X$ align with human commonsense?" For example, an image of a cat with two tails would violate commonsense. (3) Instruction Following (IF). Evaluators are asked: "Does $I_{out}^X$ follow $T$ (or $T$ and $I_0$ in the case of TI2I)?" All questions in the evaluation process are binary: answers must be either "Yes" or "No". A "Yes" earns 1 point; "No" earns 0. An image scoring 5 points is considered correct in overall. Note that throughout the human evaluation process, model $X$ remains hidden from evaluators to ensure fairness. Each record is evaluated by three different annotators to reduce human bias. For each model, we normalize the scores for each evaluation criterion by dividing the total score for that criterion by the number of samples. The resulting value is used as the final score for that evaluation criterion.

Table 1: Comparison with SOTA image generation models on Img-Ref-T2I, where "Raw" represents the model without our framework, "Ours" stands for incorporating the Raw baseline into our framework. "N/A"/"GT"/"Search" denote use "nothing"/"ground-truth"/"our frame's" reference images. The value before/after "/" indicates the score on the T2I/TI2I task.

| Type | Model | Variant | Reference Image | | | AQ | | | CC | IF | Overall |
|------|-------|---------|-----|-----|--------|--------|-------|-----------|-----|-----|---------|
| | | | N/A | GT | Search | Layout | Color | Clearness | | | |
| T2I | FLUX | Raw | ✓ | - | - | 95.3/- | 93.3/- | 87.3/- | 91.3/- | 3.3/- | 3.3/- |
| | DDPM | Raw | - | ✓ | - | 77.2/- | 84.8/- | 48.7/- | 81.6/- | 72.1/- | 40.5/- |
| TI2I | Step1X-Edit | Raw | ✓ | - | - | -/71.9 | -/85.0 | -/48.8 | -/80.6 | -/5.0 | -/3.1 |
| Omni. | Gemini | Raw | ✓ | - | - | 96.3/91.2 | 97.5/96.0 | 88.2/89.3 | 91.9/91.9 | 12.9/9.4 | 11.1/6.7 |
| | | Raw | - | ✓ | - | 93.7/93.0 | 96.2/95.5 | 81.8/87.3 | 92.5/95.0 | 67.3/31.2 | 56.6/26.1 |
| | | **Ours** | - | - | ✓ | 95.8/87.7 | 97.0/96.7 | 90.4/83.2 | 94.6/89.7 | 52.1/13.5 | 45.5/10.3 |
| | GPT-4o | Raw | ✓ | - | - | 98.0/96.3 | 99.3/100 | 98.0/98.1 | 97.4/94.4 | 20.4/33.3 | 19.1/32.7 |
| | | Raw | - | ✓ | - | 99.3/97.4 | 100/100 | 98.5/96.8 | 96.3/96.8 | 69.9/76.0 | 65.4/72.7 |
| | | **Ours** | - | - | ✓ | 98.6/98.6 | 99.3/100 | 93.7/95.7 | 95.8/96.4 | 55.2/61.2 | 48.3/57.6 |

Experimental results are listed in Table 1. We can observe that: (1) On our proposed Img-Ref-T2I dataset, existing models struggle to generate correct images, indicating that knowledge uncertainty significantly impacts the robustness of image generation. (2) Even when using ground-truth GT reference images as context, the generation accuracy of Gemini and GPT-4o still leaves room for improvement. (3) By leveraging the proposed Search-T2I framework, Gemini and GPT-4o achieve performance comparable to that with GT reference images, demonstrating the effectiveness of our framework in selecting appropriate reference images.

**Text Prompt without Uncertain Knowledge.** For commonly-used prompts that do not involve uncertain knowledge, we conduct experiments on a subset of GenEval Ghosh et al. (2023) and ConceptMix Wu et al. (2024b), as shown in Table 2. For ConceptMix, $k$ indicates the complexity of the prompt — the larger the $k$, the more complex the prompt. (1) On such datasets, the probability of the active retrieval (AR) module concluding that "retrieval is unnecessary" exceeds 95%. In such cases, our framework does not invoke retrieval, and the performance converges to that of the baseline. (2) To validate the harmlessness of our framework, we removed the AR module, and the results show that our framework can still bring performance improvements. (3) We remove the self-reflection (SR) module, and experimental results show the performance decreases, which demonstrate the effectiveness of the self-reflection module on commonly-used prompts.

Table 2: Comparison on commonly-used prompts.

| Model | Variant | AR | SR | GenEval | ConceptMix | | |
|-------|---------|----|----|---------|-----|-----|-----|
| | | | | | k=1 | k=4 | k=7 |
| GPT-4o | Raw | - | - | 92 | 92 | 46 | 30 |
| | **Ours** | ✓ | - | 92 | 95 | 52 | 30 |
| | | - | ✓ | 92 | 96 | 54 | 34 |
| | | ✓ | ✓ | 92 | 96 | 52 | 32 |

## 5.3 Preference Evaluation

Preference evaluation refers to the process of comparing two images, $I_{out}^{X_1}$ and $I_{out}^{X_2}$, generated by models $X_1$ and $X_2$ respectively for a given text prompt $T$. By assessing $I_{out}^{X_1}$ and $I_{out}^{X_2}$ from multiple dimensions, we determine which model performs better.

**Human Preference Evaluation.** Similar to Section 5.2, we begin the human evaluation by comparing $I_{out}^{X_1}$ and $I_{out}^{X_2}$ from three aspects: aesthetic quality, commonsense consistency, and instruction alignment. However, unlike Section 5.2, the evaluation questions shift from "Is it . . . ?" to "Which model, $X_1$ or $X_2$, performs better?" In addition, we introduce an extra question: "Considering all the above aspects, the image generated by which model is more suitable as the final generated result?" to evaluate the overall preference. All questions are designed to have binary answers, either "$X_1$" or "$X_2$", to prevent ambiguous or indecisive responses from evaluators.

We conduct the human preference evaluation by comparing the baseline models with their counterparts integrated into our proposed framework, $i.e.$, $X_1$ and $X_2$, respectively. Two sets of evaluations were

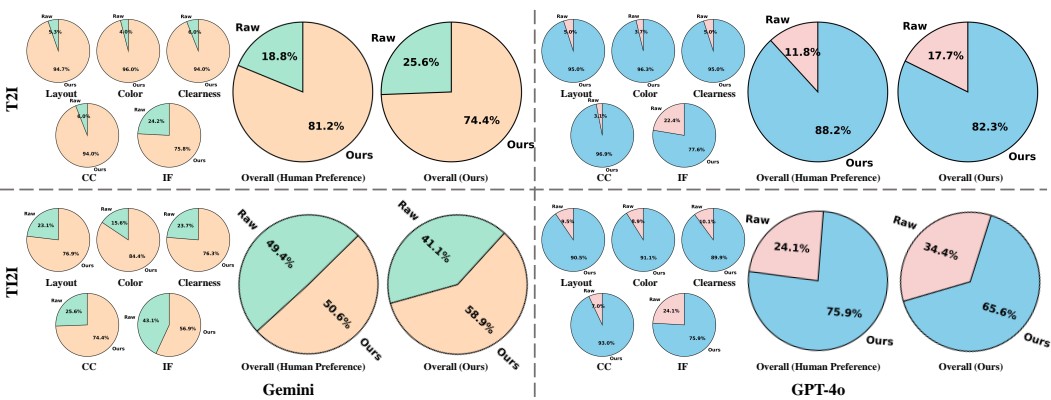

Figure 3: Experimental results of preference evaluation.

Table 3: Ablation studies of our framework on Img-Ref-T2I.

| Model | Variant | Query Generator | | | Diversity Selection | Re-Rank | | | Self Reflection | Acc. | |
|---|---|---|---|---|---|---|---|---|---|---|---|
| | | Ori. | Qwen2.5-VL | GPT-4o | | Qwen2.5-VL | GPT-4o | Human | | T2I | TI2I |
| | Raw | - | - | - | - | - | - | - | - | 19.1 | 32.7 |
| GPT-4o | **Ours** | ✓ | - | - | - | - | - | ✓ | - | 51.4 | 61.2 |
| | | - | ✓ | - | - | - | - | ✓ | - | 56.1 | 67.6 |
| | | - | - | ✓ | - | - | - | ✓ | - | 58.0 | 69.7 |
| | | - | - | ✓ | - | ✓ | - | - | - | 25.8 | 39.8 |
| | | - | - | ✓ | - | - | ✓ | - | - | 26.9 | 40.2 |
| | | - | - | ✓ | ✓ | - | ✓ | - | - | 40.8 | 49.4 |
| | | - | - | ✓ | ✓ | - | ✓ | - | ✓ | **48.3** | **57.6** |

performed, with the baseline models being Gemini and GPT-4o. The experimental results are presented in Figure 3. It can be observed that, for both the T2I and TI2I tasks, human evaluators consistently prefer the images generated using our framework. This indicates that the proposed framework significantly improves the performance of the baseline models.

**GPT-4o Preference Evaluation.** Conducting large-scale human preference evaluations is extremely costly. To address this, we design a complex prompt to guide GPT-4o in making preference evaluation based on GPT-4o, which enables automatic assessment and significantly reduces human labor costs. For the T2I task, GPT-4o preference evaluation relies on the input set $[T, I_1, I_2, I_{ref}]$. For the TI2I task, it additionally considers the initial image $I_0$. The prompt is provided in the Appendix. Experimental results of this automated preference evaluation are denoted as Ours and shown in Figure 3, demonstrating that the scores produced by our pipeline are comparable to those from the human preference evaluation, highlighting the potential of automatic preference evaluation.

## 5.4 Ablation Studies

The results of the ablation study are shown in Table 3, where "Acc." denotes the overall score of the generated images as assessed by human evaluation. We incrementally integrate different components of our proposed framework on top of GPT-4o, and explored several implementation variants for specific modules. The following observations can be made: (1) The framework is not highly sensitive to the choice of query generator. Using the original text prompt as the query (Ori.) or extracting queries via Qwen2.5-VL Bai et al. (2025) yields competitive performance. (2) When reference images are manually selected by humans (Human), the generated images are still not entirely accurate, indicating that T2I and TI2I models require further improvements in utilizing reference images effectively. (3) Performing an initial filtering of search engine results using image clustering significantly enhances generation accuracy by reducing the difficulty of the re-ranking process. (4) Incorporating a self-reflection mechanism further improves generation accuracy. These findings demonstrate that the components of our framework are both effective and complementary.

We perform an ablation study on the criteria used in the self-reflection mechanism, as shown in Table 4. $(I_i^o, T)$, $(I_i^o, I_i^{ref})$, and $(I_i^o, I_{i-1}^o)$ correspond to criteria 1, 2, and 3 in Section 3.6, respectively. The results indicate that employing all three criteria together leads to higher image generation accuracy, demonstrating the complementarity of the criteria and the rationality/effectiveness of the self-reflection mechanism.

Table 4: Ablation studies of self-reflection.

| Model | Variant | $(I_i^o, T)$ | $(I_i^o, I_i^{ref})$ | $(I_i^o, I_{i-1}^o)$ | T2I | TI2I |
|---|---|---|---|---|---|---|
| GPT-4o | Raw | - | - | - | 19.1 | 32.7 |
| | **Ours** | ✓ | - | - | 41.7 | 46.6 |
| | | ✓ | ✓ | - | 43.6 | 51.2 |
| | | ✓ | ✓ | ✓ | **48.3** | **57.6** |

## 5.5 ANALYSIS OF ACTIVE RETRIEVAL

For the two distinct lines of the active retrieval module mentioned in Sec. 5.5, the experimental results using different prompts are shown in Table 5. In this table, "Dependency" indicates which inputs a given prompt relies on to determine whether to trigger active retrieval. For the details of each prompt, please refer to Figure 7 provided in the **Appendix**. Specifically, $T$, $I^o$, and $I_0$ represent the text prompt, the image output by the model without reference images, and the original image (only applicable to TI2I), respectively. "Acc." is the accuracy of the active retrieval module, defined as the percentage of correct decisions. We observe that: (1) For T2I, using more dependencies and providing more detailed prompts can lead to higher accuracy. (2) For TI2I, relying solely on the text prompt $T$ for judging actually yields better results. This implies that dependencies and prompts should be adjusted according to the specific task.

Table 5: Analysis of active retrieval.

| Task | Prompt | Dependency | | | Acc. |
|---|---|---|---|---|---|
| | | $T$ | $I^o$ | $I_0$ | |
| T2I | prompt1 | ✓ | - | - | 90.0 |
| | prompt2 | ✓ | - | - | 80.8 |
| | prompt3 | ✓ | ✓ | - | **95.8** |
| TI2I | prompt6 | ✓ | - | - | **91.7** |
| | prompt4 | ✓ | ✓ | - | 74.2 |
| | prompt5 | ✓ | ✓ | ✓ | 80.8 |

## 5.6 ANALYSIS OF SEARCH ENGINE

We conduct ablation studies on the use of search engines, and the results are presented in Table 6, where we use GPT-4o as the baseline. We can observe that: (1) Calling the search engine at different times can affect the performance of our framework, but the impact is minimal and within an acceptable range. (2) The performance varies across different search engines, likely due to differences in the quality and relevance of the returned reference images. (3) Using multiple search engines simultaneously can yield the best performance, but it also introduces higher time costs and increased API usage, requiring a trade-off between performance and efficiency.

Table 6: Analysis of search engine.

| Search Engine | Search Time | T2I | TI2I |
|---|---|---|---|
| Bing | April | 48.3 | 57.6 |
| Bing | July | 47.4 | 56.4 |
| Google | July | 51.7 | 55.8 |
| Bing+Google | July | 53.1 | 57.7 |

## 5.7 ANALYSIS OF DIVERSITY SELECTION

This section investigates the necessity of diversity selection. We compared the performance of using different numbers of clusters for diversity selection with a baseline where all images retrieved are directly used for Re-Rank. The experimental results are shown in Figure 4. "Acc." represents the ratio of cases where the Top-1 image obtained through Re-Rank is a suitable reference image, as judged by humans. "C($x$)" indicates the use of diversity selection with $x$ clusters, while "NC" means no diversity selection is applied. From the figure, the following observations can be made: (1) When the number of clusters is less than 10, as the number of clusters increases, the ratio of correctly ranked Top-1 images also increases. (2) When the number of clusters exceeds 10, the correct Re-Rank rate begins to decrease as the number of clusters increases. (3) When no diversity selection is used, the accuracy of Re-Rank is the lowest, because an excessive number of images introduces noise into the Re-Rank process. These observations demonstrate that diversity selection is both necessary and effective.

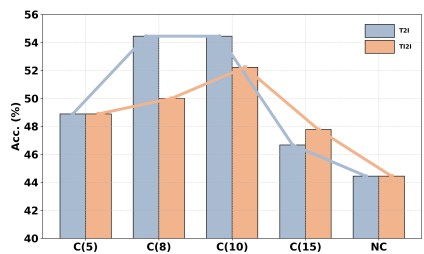

Figure 4: Analysis of diversity selection.

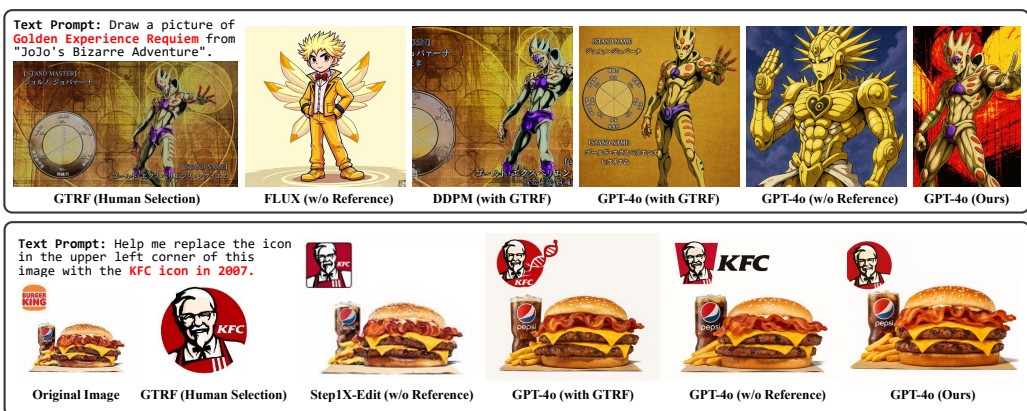

Figure 5: Qualitative comparisons on the proposed Img-Ref-T2I dataset, where GTRF denotes the ground-truth reference image provided in Img-Ref-T2I by human selection.

## 5.8 ANALYSIS OF TIME/COMPUTE OVERHEAD

We measure the average runtime and number of API calls for each component of our framework, as shown in Table 7. For common prompts, our framework introduces almost no additional computation time or resource consumption, as demonstrated in Sec. 5.2. For prompts involving uncertain knowledge, the framework introduces approximately 22.4 seconds, 3 search engine calls, 5.2 GPT-4o API calls in average. Our work can be viewed as a deep retrieval framework for T2I generation, such frameworks are typically time-consuming,

Table 7: Analysis of overhead.

| Component | Call | Runtime | Times |
|---|---|---|---|
| Active Retrieval | GPT API | 0.6 | 1 |
| Query Generator | GPT API | 0.5 | 1 |
| Search Engine | Bing Search | 2.2 | 3 |
| Diversity Selection | GPT API | 2.2 | 1.6 |
| Re-Rank | GPT API | 6.2 | 1.6 |
| Self-Reflection | GPT API | 0.8 | 1.6 |

often adding several minutes of overhead. In contrast, the additional time of our framework is acceptable. Furthermore, we provide a fully open-source-based framework (excluding the search engine) via Qwen2.5-VL, which significantly reduces resource consumption compared to using the GPT API, while still achieving comparable results, whose details are provided in the **Appendix**.

## 5.9 QUALITATIVE ANALYSIS

Figure 5 depicts qualitative examples from the Img-Ref-T2I dataset. The words in bold red denotes a short description of the ground-truth reference image (GTRF). We can observe that: (1) GPT-4o (Ours) performs the best, consistently generating reasonable images. (2) GPT-4o (with GTRF) performs slightly worse than GPT-4o (Ours). Due to the availability of only one GTRF and the lack of a self-reflection process, the generated images often contain minor issues. For example, in the second example, the icon generated by GPT-4o (with GTRF) has noticeable flaws. (3) DDPM uses the GTRF but merely imitates it without deeper understanding. (4) FLUX and Step1X-Edit cannot utilize any reference images and perform poorly. These observations demonstrate that Search-T2I is effective and, in some scenarios, can even produce results that are better than the GTRF.

## 6 CONCLUSION

In this work, we have presented Search-T2I, a framework that integrates reference images from the Internet into T2I/TI2I models, which can effectively mitigate inaccurate image generation caused by uncertain knowledge in text prompt. An Img-Ref-T2I dataset that includes three types of scenarios involving uncertain knowledge is curated by human experts. The dataset enables evaluate the ability of T2I/TI2I models to generate images when the textual knowledge is uncertain. Experimental results on Img-Ref-T2I demonstrate that our framework can significantly enhances the image generation performance of T2I/TI2I models in these challenging scenarios.

**Reproducibility Statement.** We have made efforts in the following aspects to ensure the reproducibility of the paper: (1) The code and data will be open-sourced after the paper is accepted. (2) We conduct experiments on different types of models (including both open-source and closed-source models) to validate the general applicability of the proposed framework. (3) We repeat the same experiments at different times to demonstrate that the framework is time-insensitive. (4) We provide a framework and evaluation methods based on open-source models, significantly improving reproducibility.

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

## A APPENDIX

### A.1 PROMPTS

In this section, we provide the prompts of this paper, as shown in Figure 6. In addition, the prompt candidates for active retrieval are shown in Figure 7.

### A.2 FULLY OPEN-SOURCE-BASED FRAMEWORK

We implement a fully open-source-based framework (excluding the search engine) via Qwen2.5-VL Bai et al. (2025), to avoid introducing additional resource consumption (*e.g.*, GPT API calls). The experimental results using Qwen2.5-VL as the component of our framework are shown in the Table 8. We can observe that both Qwen2.5-VL and GPT-4o deliver comparable performance.

Table 8: Comparison on different components.

| Model | Variant | Component | | T2I | TI2I |
| | | GPT-4o | Qwen2.5-VL | | |
|---|---|---|---|---|---|
| GPT-4o | Raw | - | - | 19.1 | 32.7 |
| | **Ours** | - | ✓ | 44.2 | 53.4 |
| | | ✓ | - | 48.3 | 57.6 |

**Prompt for Query Generator:**
Here is a problem of drawing a picture:{question}. This problem is usually a request to draw an object. You need to condense this problem into a search term so that I can search the Internet for relevant pictures to assist in the drawing of the picture. Finally, you only need to return the query term you have extracted, the language of the search term you return should be consistent with the language of the question (Chinese, Japanese, English), without any additional output.

**Prompt for Re-Rank:**
Here is an image generation/editing request (Chinese/Japanese/English): {prompt}. There are {len(imgs)} reference images named {', '.join(names)}.
These problems are all about object drawing. You need to identify the drawn object from three languages. You don't need to output this object to me. Instead, you only need to rank the degree of match between these pictures and the object. You don't need to consider other factors such as image quality and aesthetics. You only need to focus on the correspondence between the image content and the drawn object in the question. Rank them from the highest to the lowest degree of match, and only reply with the sorted picture numbers, for example, '2 3 1....'. All the input pictures need to be sorted.

**Prompt for Self-Reflection:**
**(Criteria 1)** Does the generated image closely follow the text instructions? You need to break down the text problem, analyze the specific content and some characteristics of the object to be drawn. Based on these summarized characteristics, you evaluate the quality of the generated image. Score it according to the correspondence between the content and characteristics in the image. If it corresponds well, you can give 2 points. If only part of it corresponds, give 1 point. If there is no correspondence at all, give 0 points.

**(Criteria 2)** Whether it conforms to the correct knowledge, you need to judge whether the content of the reference picture can assist in the drawing of this drawing/editing problem based on this problem. Because the reference picture may be wrong, having no relation to the problem at all, or there may be a certain correlation between the reference picture and the problem, but there are deviations in some details. You rate it according to the degree of matching. The content of the reference picture fully meets the requirements of the problem: 4 points. The content of the reference picture partially meets the requirements of the problem: 2 points. Completely wrong: 0 points.

**(Criteria 3)** Is there a reference relationship between the generated image and the reference image? The first image is the generated one, and the second is the reference image. You need to analyze and compare the main content of the two pictures, mainly including objects, colors, environment, proportion, posture, etc. If the reference relationship between the two pictures is strong and the main body in the pictures is basically the same, you can give 2 points. If there is only partial reference, such as color, appearance, etc., you can give 1 point. If there is no reference relationship at all, give 0 points.

**(Criteria 4)** Now you are a data annotator without any bias. You must carefully and conscientiously label each piece of my data according to the following requirements. My data is the scoring of the quality of some text-to-image questions. This is an image drawing question {prompt}. The first picture is a new picture drawn based on this question, and the second picture is another answer to this question. You now need to compare the quality of these two pictures. If you think the first picture is closer to this question, you output "True", otherwise you output "False". Your final answer can only be "True" or "False", without any other redundant output.

**Prompt for GPT-4o Preference Evaluation:**
**(For T2I)** Now you are a data annotator without any bias. You must carefully and earnestly label each piece of my data, making a binary classification choice for each piece of data. You must make a choice and cannot refuse to answer.This is a question about drawing: {prompt}. The first image is the reference image. The second image is generated based on this question and has no necessary connection with the reference image. The third image is also generated based on this question and has no necessary connection with the reference image either.However, this reference image is a necessary element in answering this image question. You need to evaluate the quality of the next two images based on the text question and the reference image. Which model - generated image is more suitable as the final generated result? To evaluate the overall preference. All questions are designed to have binary answers.If the second image is closer to the reference image, you output 0. If the third image is closer to the reference image, you output 1. Your final output is only allowed to be 0 or 1, without any other extra output. You are not allowed to output any other language except 0 and 1, and you cannot refuse to answer.

**(For TI2I)** Now you are a data annotator without any bias. You must carefully and conscientiously label each piece of my data. Make a binary classification choice for each piece of data. You must make a choice and cannot refuse to answer. This is a question about image drawing: {prompt}. The first image is the reference image. The second image is generated by editing the second image according to this question. There is no inevitable connection between it and the reference image. The third image is also generated by editing the second image according to this question. There is no inevitable connection between it and the reference image, either.However, this reference image is a necessary element in answering this image question. You need to evaluate the quality of the next two images based on the text question, the reference image, and the edited images. Which model-generated image is more suitable as the final generated result? To evaluate the overall preference. All questions are designed to have binary answers.If the third image is closer to the reference image, you output 0. If the fourth image is closer to the reference image, you output 1. Your final output is only allowed to be 0 or 1. Do not have any other redundant output. You are not allowed to output any other language except 0 and 1, and you cannot refuse to answer.

Figure 6: The prompts used in this paper.

**Prompt1:** Analyze the following image-generation prompt and decide whether it necessitates querying a search engine for external knowledge. Answer a single letter 'Y' or 'N'.

Prompt to evaluate: {}
---
Examples:
1. "Draw a picture of Golden Experience Requiem from \"JoJo's Bizarre Adventure.\"" → Y
2. "Draw a picture: the Statue of Liberty was transported on the ship." → N
3. "Dress Mao Mao from Boonie Bears in the armor from the picture." → Y
4. "Help me draw a picture: the Sphinx under the direct sunlight." → N

**Prompt2:** Task: Given an image-generation prompt, decide if it requires external knowledge from a search engine to generate accurately. Answer strictly single letter "Y" or "N" based on these rules: Prompt to evaluate: {}

**Prompt3:** Given a text-to-image generation prompt and a generated image, strictly determine if they match. Respond with single letter Y or N. Prompt: {}

**Prompt4:** Given a image edition prompt and a edited image, strictly determine if they match. Respond with single letter Y or N. Prompt: {}

**Prompt5:** Given a image edition prompt, a original image (the first) and a edited image (the second), strictly determine if they match. Respond with single letter Y or N. Prompt: {}

**Prompt6:** Analyze the following image-edition prompt and decide whether it necessitates querying a search engine for external knowledge. Answer a single letter 'Y' or 'N'.

Prompt to evaluate: {}
---
Examples:
1. "Replace the character in the picture with the sea yaksha from \"Nezha: The Devil Child Comes to the Sea.\"" → Y
2. "Let the person in the picture make a phone call with a Huawei Mate70." → Y
3. "Add a bird of another color to the picture." → N
4. "Help me replace the wheels of the car in the picture with tracks." → N

Figure 7: Prompt candidates for active retrieval.

Furthermore, we perform experiments using Qwen2.5-VL for evaluation, rather than GPT-4o and Human, to enhances the reproducibility of our framework. Experimental results are shown in the Table 9. It can be seen that Qwen achieves performance comparable to GPT-4o, which enhances the reproducibility of our framework.

### A.3 USE OF LARGE LANGUAGE MODELS

In this paper, two aspects of large language models were used: (1) We use prompts like "Optimize my phrasing: [*my content*]" to leverage GPT for improving the wording. (2) We use the prompt"If there are any grammatical errors in the following content, please point them out: [*my content*]" to utilize GPT for avoiding grammatical mistakes.

Table 9: Comparison of using different evaluators.

| Model | Evaluator | Vote (T2I) | | Vote (TI2I) | |
|---|---|---|---|---|---|
| | | Raw | Ours | Raw | Ours |
| Gemini | Human | 19 | 81 | 49 | 51 |
| | GPT-4o | 26 | 74 | 41 | 59 |
| | Qwen2.5-VL | 28 | 72 | 41 | 59 |
| GPT-4o | Human | 12 | 88 | 24 | 76 |
| | GPT-4o | 18 | 82 | 34 | 66 |
| | Qwen2.5-VL | 23 | 77 | 30 | 70 |

## A.4 LIMITATION

The Img-Ref-T2I dataset constructed in this paper relies on manual collection and annotation, making large-scale expansion extremely costly. In the future, we plan to explore automatic collection and annotation of T2I and TI2I samples with uncertain knowledge.