# OpenReview forum: "Search-T2I: Internet-Augmented Text-to-Image Generation"
_ICLR.cc/2026/Conference — Submitted to ICLR 2026_

### Official Review · Reviewer_pE1r · 2025-10-28

**Soundness:** 3
**Presentation:** 3
**Contribution:** 1
**Rating:** 4
**Confidence:** 5

**Summary:**

This summary introduces Search-T2I, a framework designed to improve text-to-image models to generate state-of-the-art images by retrieval augmentation. The system uses an internet search to find relevant reference images, actively decides when they are needed, selects the most suitable ones, and employs a self-reflection mechanism to refine the output. The framework is tested on a dedicated dataset with various uncertainty types, and significantly outperforms a model like GPT-4o in human evaluations.

**Strengths:**

1. This paper introduces a internet-augmented framework that actively retrieves and utilizes reference images, moving beyond standard T2I models to dynamically resolve knowledge uncertainty.
2. The framework demonstrates superior performance, significantly outperforming strong baselines like GPT-4o by a large margin in human evaluation.
3. The paper is clearly structured, logically explaining the problem and systematically introducing each component of the solution, which makes the framework easy to understand.

**Weaknesses:**

1. This work is based on prompting, and the focus should be on pipeline design. However, the overall pipeline of this paper is trivial, where all modules, including the query reformulation module, retrieval module, reflection module, etc., have been widely used in previous works such as [1]. This significantly diminishes the paper's innovation.
2. There is a lack of comparison with existing retrieval-augmented image generation works, such as [1][2][3][4], making it impossible to assess the effectiveness of the proposed framework.

[1]. Huaying Yuan, Ziliang Zhao, Shuting Wang, Shitao Xiao, Minheng Ni, Zheng Liu, and Zhicheng Dou. Finerag: Fine-grained retrieval-augmented text-to-image generation.
[2]. Wenhu Chen, Hexiang Hu, Chitwan Saharia, and William W Cohen. Re-imagen: Retrieval-augmented text-to-image generator.
[3]. Andreas Blattmann, Robin Rombach, Kaan Oktay, Jonas Müller, and Björn Ommer. Retrieval-augmented diffusion models.
[4]. Rotem Shalev-Arkushin, Rinon Gal, Amit H. Bermano, Ohad Fried. ImageRAG: Dynamic Image Retrieval for Reference-Guided Image Generation.

**Questions:**

Same to the weaknesses.

---

### Official Review · Reviewer_LVS4 · 2025-10-29

**Soundness:** 2
**Presentation:** 2
**Contribution:** 2
**Rating:** 2
**Confidence:** 4

**Summary:**

The papaer propose an Internet-Augmented text-to-image generation framework (Search-T2I) to compel T2I models clear about such uncertain knowledge by providing them with reference images. Experiments prove the effectiveness of the proposed framework.

**Strengths:**

Pros:
1. The idea is interesting, using internet to augment exisiting generation model.
2. Compared against GPT-4o, the performance is good.

**Weaknesses:**

1. I have come across a paper in Arxiv: IA-T2I: Internet-Augmented Text-to-Image Generation (Chuanhao Li, et, al.); which has some overlap with this submission, while different titles and some details. Is this the same paper? If not, I do not find any citations.

2. The citations should be regulated. For instance, FLUX Labs (2024). What labs? You should specify BFL.

3. There is no ethnic approvement for using human sources to evaluate. Does they work fairly? Is there any forced labor happens?

4. The collected dataset Img-Ref-T2I is not public available.

5. The proposed Search-T2I framework is a combination of existing works. No novel.

**Questions:**

NA

**Details Of Ethics Concerns:**

1. The paper uses Internet data without specific web crawling policies
2. The paper does not mention annotator compensation, which may cause forced labor.
3. The paper has significant overlap with a paper on Arxiv with a different title.

---

### Official Review · Reviewer_Mjnz · 2025-10-30

**Soundness:** 3
**Presentation:** 2
**Contribution:** 2
**Rating:** 6
**Confidence:** 4

**Summary:**

The paper presents a training-free framework that actively retrieves and hierarchically filters reference images from the internet to augment text-to-image (T2I) and text-driven image editing (TI2I) models on-the-fly, mitigating generation errors caused by uncertain knowledge.

**Strengths:**

1. From the perspective of design motivation, the paper's focus on a training-free framework, which  demonstrates notable advantages in both T2I and TI2I applications, offering valuable insights for future research.

2. By extending the IAG techniques traditionally used in LLMs to the image-generation domain and providing a well-designed diversity-based selection strategy, the work represents an effective and welcome technological expansion.

**Weaknesses:**

- Section 3.6: The section say “three key criteria” but lists four; criteria (2) and (3) appear to overlap. Table 4 further claims that criteria (1)(2)(3) were used, yet the correct set should be (1)(2)(4). Please verify whether (2) and (3) were originally intended as a single criteria.

- Section 5.2: The evaluation protocol does not detail reliability or bias-mitigation steps. Clarify how personal preference differences of the (co-author) evaluators influent the evaluation and how the influence were controlled.

- Table 2 in Section 5.2: Removing the Active Retrieval (AR) module still improves scores, even outperforming the full framework. Discuss why retrieved information (AR + SR) degrades performance versus self-reflection alone; a brief ablation or error analysis is expected.

- Table 5: The distinctions between prompts are relegated to the appendix, leaving the table itself uninformative. Add a concise in-text summary of how the prompts differ.

- Table 6: Google April data are missing, precluding a clear time-comparison; the text can only assert that multi-engine retrieval helps. Moreover, July data (later and should be more abundant) yield lower scores than April—an unexpected trend. Provide a brief analysis for why performance drops with more information or what difference between multi-engine retrieval and later retrieval.

**Questions:**

Please see weakness for my questions

---

### Official Review · Reviewer_DuHG · 2025-11-01

**Soundness:** 2
**Presentation:** 2
**Contribution:** 2
**Rating:** 2
**Confidence:** 4

**Summary:**

The paper presents IA-T2I, an Internet-augmented text-to-image generation framework that addresses the limitation of T2I models when encountering text prompts containing uncertain knowledge (rare, unknown, or ambiguous information). The framework incorporates an active retrieval module to determine when reference images are needed, a hierarchical image selection process that first clusters retrieved images for diversity and then re-ranks candidates via large vision-language models, and a self-reflection mechanism that iteratively evaluates generated images against the prompt, reference image utility, and previous outputs to decide whether to accept or regenerate. To evaluate the framework, the authors construct the Img-Ref-T2I dataset comprising T2I and TI2I tasks. The author claims integrating IA-T2I into GPT-4o and Gemini can outperforming raw GPT-4o by ~30 % in human evaluation, while ablations confirm the complementary contributions of diversity selection and self-reflection.

**Strengths:**

1. IA-T2I innovatively introduces a proactive retrieval module to determine when a reference image is needed and designs a two-level selection strategy of "diversity clustering + re-ranking" to effectively filter out a large number of noisy images returned by the search engine, providing the T2I model with the most instructive single reference image and significantly improving generation accuracy in scenarios with uncertain knowledge.

2. Both human and GPT-4o evaluation are integrated in this loop.

**Weaknesses:**

1. The self-reflection loop is presented as a principal contribution, yet **iterative generate-evaluate-regenerate pipelines is common standard** in both LLM and AIGC literature since early 2024; the paper neither positions the proposed mechanism against prior reflection frameworks (e.g., Self-Refine, Iterative Prompting) nor quantifies its marginal gain over single-shot or alternative reflective baselines, rendering the claimed novelty unsubstantiated.

2. The experimental evidence relies almost exclusively on manual ratings provided by humans; objective metrics commonly adopted in T2I evaluation—CLIP-Score, FID, DSG, etc.—are absent, and inter-annotator agreement is not reported, leaving the results vulnerable to confirmation bias and statistical insignificance.

3. **The adoption of GPT-4o as an automatic preference evaluator is elevated to a core contribution; however, LLM-as-a-judge is already a widespread practice in generative vision research.** The authors offer no correlation analysis (Pearson/Spearman), significance tests, or comparisons with existing learned metrics (HPSv2, ImageReward, PickScore), overstating the methodological advance.

4. Img-Ref-T2I comprises **only 240 human-curated instances**; this scale is insufficient to stabilise performance estimates, and no power analysis is supplied to justify that the dataset can reliably detect the hypothesised effect size.

5. Prompts in the dataset average fewer than 20 words and exhibit limited compositional complexity; consequently, model scores cluster narrowly, failing to expose differences under the long-form, multi-attribute, or combinatorial instructions emphasised by contemporary benchmarks such as DrawBench or T2I-CompBench.

6. The baseline pool omits several competitive T2I systems (e.g., Hunyuan, Doubao) that may exhibit distinct behaviours under the proposed retrieval-augmented regime, thereby constraining the external validity of the comparative findings.

7. **The paper asserts “high consistency” between GPT-4o-based automatic ratings and human preferences, yet provides no quantitative corroboration**—neither correlation coefficients nor error breakdowns. Its strongly recommended to read some established protocols like **T2I-CompBench, T2V-CompBench or V-Bench** that mandate Pearson r ≥ 0.8 and detailed failure-mode analysis; without this verification, all subsequent claims of performance improvement rest on an unvalidated evaluator and remain scientifically fragile.

**Questions:**

I believe this paper needs a complete overhaul, particularly regarding the weaknesses section. Compared to ImageReward and Pick-a-Pic, which are two years old, it offers little advantage, let alone compared to more recent papers. Furthermore, the dataset is far too small; no AIGC dataset has ever been this small. Additionally, the authors should have avoided exaggerating their contributions.

---

### Meta-Review · Area_Chair_AMiG · 2025-12-25

**Summary:**

This paper proposes Search-T2I, an internet-augmented text-to-image generation framework designed to improve performance when prompts contain uncertain knowledge (e.g., rare, unknown, or ambiguous information). The framework aims to mitigate generation errors by providing the model with reference images retrieved from the internet. While reviewers generally agreed that using the internet to augment generation models is an interesting concept, the majority (3 out of 4) recommended rejection. Their primary concerns include limited technical novelty, a lack of baseline comparisons with existing work, non-robust evaluation (e.g., relying on a small dataset), and insufficient discussion of experimental results. Consequently, the paper is recommended for rejection.

**Reviewer Concerns:**

The authors did not provide rebuttal.

**Reviewer Scores:**

The authors did not provide detailed rebuttal to each reviewer, so the reviewers were not involved in the discussion.

---

### Decision · Program_Chairs · 2026-01-26

Reject